# Extremely Ultranarrow Linewidth Based on Low-Symmetry Al Nanoellipse Metasurface

**DOI:** 10.3390/nano13010092

**Published:** 2022-12-24

**Authors:** Liangyu Wang, Hong Li, Jie Zheng, Ling Li

**Affiliations:** Laboratory of Micro-Nano Optics, School of Physics and Electronic Engineering, Sichuan Normal University, Chengdu 610101, China

**Keywords:** surface plasmon resonance, ultranarrow linewidth, low-symmetry Al metasurface, UV, DUV

## Abstract

Plasmonic nanostructures with ultranarrow linewidths are of great significance in numerous applications, such as optical sensing, surface-enhanced Raman scattering (SERS), and imaging. The traditional plasmonic nanostructures generally consist of gold and silver materials, which are unavailable in the ultraviolet (UV) or deep-ultraviolet (DUV) regions. However, electronic absorption bands of many important biomolecules are mostly located in the UV or DUV regions. Therefore, researchers are eager to realize ultranarrow linewidth of plasmonic nanostructures in these regions. Aluminum (Al) plasmonic nanostructures are potential candidates for realizing the ultranarrow linewidth from the DUV to the near-infrared (NIR) regions. Nevertheless, realizing ultranarrow linewidth below 5 nm remains a challenge in the UV or DUV regions for Al plasmonic nanostructures. In this study, we theoretically designed low-symmetry an Al nanoellipse metasurface on the Al substrate. An ultranarrow linewidth of 1.9 nm has been successfully obtained in the near-UV region (400 nm). Additionally, the ultranarrow linewidth has been successfully modulated to the DUV region by adjusting structural parameters. This work aims to provide a theoretical basis and prediction for the applications, such as UV sensing and UV-SERS.

## 1. Introduction

Plasmonic nanostructures with ultranarrow linewidths have aroused great attention in numerous applications, such as optical sensors [1,2,3,4,5,6], imaging [7,8], and surface-enhanced Raman spectroscopy (SERS) [9,10,11]. Conventional plasmonic nanostructures are generally fabricated by utilizing gold and silver materials. Although they exhibit excellent surface plasmon resonance (SPR) characteristics in the visible to infrared regions, they cannot operate in the deep ultraviolet (DUV) or ultraviolet (UV) regions due to the interband transition [12]. However, researchers are eager to realize ultranarrow linewidth of plasmonic nanostructures due to electronic absorption bands of many important biomolecules mostly located in the UV or DUV regions. Al is abundant on Earth. Importantly, Al plasmonic nanostructures can support SPR from the DUV to the near infrared (NIR) regions. Therefore, Al plasmonic nanostructures are outstanding candidates for investigating the characteristics of SPR in the DUV and UV regions [13]. In recent years, researchers have devoted great efforts to reducing SPR linewidths by theoretically and experimentally designing plasmonic nanostructures, such as metallic nanospheres [14], nanoparticle dimers [15], nanorods [16], nanotriangles [17], nanohole arrays [18,19,20,21], rhombohedral arrays [22], etc. However, the latest experimental results indicate that a 14-nm ultranarrow linewidth has been achieved based on Al nanostructures in the near-UV region [23]. However, it remains a challenge to realize an ultranarrow linewidth below 5 nm in the DUV or UV regions.

In this work, we theoretically designed a low-symmetry Al nanoellipse metasurface on the Al substrate. The theoretical mechanism derived suggests that plasmonic nanostructures with broken symmetries enable dark modes to couple to far-field radiation, which contributes to producing the ultranarrow linewidth. A 1.9-nm ultranarrow linewidth can be successfully obtained with optimized structural parameters in the near-UV region. Additionally, an anticipated remarkable ultranarrow linewidth of the SPR modes towards the extension of the DUV region is generally expected from the adjustment of the period. Moreover, plasmonic properties of the periodic Al nanoellipse metasurface are further explored by modulating structural parameters, polarization characteristics, and substrate. The ultranarrow linewidth presents excellent detection in tiny wavelength shift and exhibits great potential applications in Al-based optical sensors. This investigation will provide a theoretical basis and prediction for the applications, such as UV sensing and UV-SERS.

## 2. Materials and Methods

In this study, all numerical simulations are performed by finite-difference time-domain (FDTD) solutions. To be consistent with the real experimental environment, the perfectly matched layer (PML) boundary conditions are set in the z direction, and periodic boundary conditions in the x- and y-axis are adopted in the simulations. Considering calculation convergence, the simulation time is set as 1000 fs in all simulations. To save calculation time and computation resources, the non-uniform mesh method is adopted, and a uniform mesh size of 2 nm (x, y, and z directions) is used. The simulated model consists of an Al nanoellipse metasurface located on the Al substrate, as shown in Figure 1a. The wavelength of the Al nanoellipse metasurface is from 200 nm to 1200 nm, and the refractive index of Al comes from Palik [24]. This low-symmetry metasurface can be fabricated by laser interference lithography, nanoimprint lithography, or electron beam lithography [25,26,27]. The structural parameters of the Al nanoellipse metasurface are set as the period of 80 nm (Px) by 400 nm (Py), the semi-axis of 20 nm (minor axis, labeled as b) by 40 nm (major axis, labeled as a) in the x- and y-axis, respectively, and the height of 70 nm in the z-axis. Al film with a 100 nm thickness is selected as the substrate due to its higher reflection and smaller skin depth [18]. Measurement of the reflectance spectra is performed to investigate the optical characteristic of the Al nanoellipse metasurface under the normal incidence. Taking the polarization dependence into account, the polarization is set along the y-axis as depicted in Figure 1a, and the incident electric field amplitude is set as 1 V/m. With optimized structural parameters and the period, we successfully obtained an ultranarrow linewidth of 1.9 nm in the near-UV region (400.8 nm), as illustrated in the Figure 1b. The near-field distribution of resonant wavelength corresponding to the ultranarrow linewidth has been listed in the inset of Figure 1b. The results demonstrate that the extremely ultranarrow linewidth originates from the contribution of the surface plasmon polariton (SPP) induced by the y-axis period (Py = 400 nm) [23,28,29,30].

## 3. Results

### 3.1. Effects of the Major and Minor Axes of the Al Nanoellipse Metasurface

To clarify the mechanism of ultranarrow linewidth, we numerically calculated the far-field scattering (defined as cross sections) spectra of individual nanoparticles and the reflectance spectra of periodic arrays. For individual nanoparticles, there are two sharp peaks displayed in the corresponding scattering spectra with the increase in the semi-major axis (labeled as a) and the maintenance of the semi-minor axis (labeled as b) constant, as shown in Figure 2a. The scattering spectra of individual nanoparticles are insensitive under varied a because of the polarization along the minor axis. Keeping a constant, when b is smaller than 40 nm, there is one distinctly sharp peak displayed in the reflectance spectra. The near-field distribution corresponding to the wavelength of 369 nm is displayed in the inset, and obviously, it can contribute to dipole modes corresponding to the localized surface plasmon (SPR) modes (see Figure 2b). With the increase in b, we can observe two peaks in these wavelength regions. Besides the dipole mode, the quadrupole emerges. The near-field distribution corresponding to the wavelength of 238 nm is displayed in the inset of Figure 2b. The resonant peaks present broad linewidth (>100 nm) due to the radiative damping and dynamic depolarization. For periodic arrays, the Al nanoellipse metasurface has kept constants of 200 nm by 400 nm in the x- and y-axis, respectively, and the semi-minor axis constant. The reflectance spectra of period arrays did not show obvious change, which is similar to the results of individual nanoparticles in Figure 2a. Subsequently, keeping the semi-major axis constant, an extremely ultranarrow linewidth of 0.9 nm (corresponding to the semi-minor axis of 20 nm) can be observed as indicated in Figure 2d by appropriately modulating the semi-minor axis from 20 nm to 70 nm. When nanoparticles are arranged periodically, the large-range coupling will occur between individual nanoparticles and realize an ultranarrow linewidth [14]. The simulation results further demonstrate that the modulation of ultranarrow linewidth is mainly related to the choice of polarization direction and is insensitive to the variation of Al film thickness (see Appendix A). Thus, the optimal configuration adopts the semi-axis of 20 nm by 40 nm in the x- and y-axis, respectively, and the height of 70 nm in the z-axis.

### 3.2. Effects of Substrate and Polarization Angle Dependence

Without considering the substrate, a 0.9-nm ultranarrow linewidth of reflectance spectrum with the semi-axis of 20 nm by 100 nm in the x- and y-axis, respectively, and the height of 80 nm in the z-axis can be obtained in the near-UV region (400 nm), as shown in Figure 2d. In Figure 2c,d, when the optimal configuration adopts the semi-axis of 20 nm by 40 nm in the x- and y-axis, respectively, and the height of 70 nm in the z-axis, the reflection spectrum obtained is shown in Figure 3a (red line). The intensity of the reflection spectrum is very low, but there is still a full width at half maximum (FWHM) of 1.4 nm at 400 nm. However, when the substrate is considered, the interaction between nanoparticle and film will be dominant, and the reflectance spectrum exhibits great diversity, as displayed in Figure 3a. The interband transition can be observed at approximately an 850-nm wavelength, and the ultranarrow linewidth emerges in the DUV region (~220 nm). In order to further explore an obvious ultranarrow linewidth in the UV and DUV regions, the reflectance spectra of an Al nanoellipse metasurface on the Al substrate with different polarization angles (labeled as θ) of incident light have been recorded, as shown in Figure 3b. When θ is increased to 30°, an extremely ultranarrow linewidth of 5.5 nm appears at 404.8 nm. By continuously increasing θ, the ultranarrow linewidth becomes more obvious, and the intensity gradually enhances. A 3.9-nm ultranarrow linewidth can be successfully obtained when there is polarization along the major axis. The results further demonstrate that the ultranarrow linewidth of a low-symmetry Al nanoellipse metasurface is dependent on the polarization direction.

### 3.3. Effects of Periods in the x- and y-Axis 

Although an ultranarrow linewidth (~3.9 nm) has been acquired at 404.8 nm (corresponding to the polarization angles of 90°), this result seems unsatisfactory, and there is a stranger dip at 270 nm with a broad linewidth (>50 nm). In order to further narrow the linewidth and obtain a higher intensity dip, the periods in the x- and y-axis are taken into consideration. First, Al nanoparticles have kept constants of 40 nm and 20 nm in semi-major and semi-minor axes, respectively, with the thickness of 70 nm. The period in the y-axis (Py) is determined as 400 nm, with a varied period in the x-axis (Px) from 200 nm to 80 nm. As shown in Figure 4b, the ultranarrow linewidth of 1.9 nm is obtained with the decreasing of the period in the x-axis (Px). Moreover, an anticipated remarkable ultranarrow linewidth of the SPR modes towards the extension of the DUV region is generally expected from the adjustment of the period in the y-axis (Py). With the decrease in Py, the reflectance spectrum of the Al nanoellipse metasurface appears as a uniform blue shift from UV region to DUV region, as shown in Figure 4c. When Py gradually decreases with the change of 50 nm, the transmission dip position changes accordingly, and the change is basically consistent with the change of Py, showing a linear decreasing relationship.

### 3.4. Detection in Tiny Wavelength Shifts

The ultranarrow SPR linewidth of 1.9 nm has been successfully obtained by modulating structural parameters for an Al nanoellipse metasurface. The delicate SPR linewidth offers great possibility for metal-based sensors in the UV region. Herein, we demonstrate the reliable detection of a tiny wavelength shift by immersing an Al nanoellipse metasurface into the simulated environment with different refractive indices. The results show that the resonant modes produce slightly red shifts with the increase of the surrounding refractive index from 1.00 to 1.10 (at 0.02 intervals), and the corresponding spectral shifts are recorded in Figure 5a. A 39.815-nm wavelength shift can be observed from the small window (the inset in Figure 5a).

The evaluation of sensing typically refers to the sensitivity and figure of merit (FOM). Sensitivity is defined as S = Δλ/Δn. Here, Δλ represents the resonant dip shift. Δn denotes the change of refractive index. The corresponding sensitivity is about 398.15 nm RIU-1. The figure of merit (FOM) contributes to a more rigorous evaluation standard and is defined as S/Γ, where Γ denotes the full width at half maximum (FWHM) of corresponding resonant modes. The FOM value is about 210. Moreover, we further calculated the sensing performance of the aqueous solution with one order of magnitude lower than the change of the refractive index in Figure 5b from 1.000 to 1.014 (at 0.002 intervals). The results demonstrate that such a tiny wavelength shift can still be monitored. The sensitivity presents as 401 nm RIU-1, and the FOM is about 211. The Al nanoellipse metasurface with the ultranarrow linewidth of 1.9 nm exhibits great potential to achieve high-sensitivity UV sensing. The theoretical work will provide a fundamental theoretical analysis for the experiment in the future.

### 3.5. Experimental Results

This low-symmetry metasurface can be fabricated by laser interference lithography (LIL), nanoimprint lithography, or electron beam lithography [25,26,27]. Here, the LIL technology was adopted to fabricate the low-symmetry Al metasurface because of simplicity, economics, and reproduction. This technology works by placing the reflector in the vicinity of the sample to simplify the spatial designed and the alignment of optical path, as show in Figure 6a. The nanostructures with controllable morphology can be produced by precisely controlling exposure dose, and 1D or 2D periodic nanostructure arrays can be obtained by controlling the numbers of exposure. Mainly, a series of periodic nanostructures can be obtained by altering the incidence angle between two beams (shown in Figure 6b). The period depends on the equation $P=\frac{\lambda }{2n\sin\theta }$, where *P* denotes period, *λ* is the wavelength of incidence light, *n* is the refractive index of surrounding, and *θ* represents the incidence angles of incident light. Briefly, the positive photoresist coated onto the silicon substrates was illuminated once with an illumination angle of 55, and then the interference of the two laser beams produce one-dimensional periodic gratings. The second illumination with an illumination angle of 24 was subsequently conducted after rotating the sample 90 degrees, producing a two-dimensional ellipse metasurface. Finally, the Al film with a thickness of 80 nm was deposited onto the overall samples. The wafer-scale high-quality Al metasurface with 200 nm in Px and 400 nm in Py is shown in Figure 6c. A 12-nm linewidth has been obtained in the near-UV region with polarized light, which is same as that of the simulation, as shown in Figure 6d. The simulation results demonstrate that a 3.9-nm linewidth can be obtained (as shown in Figure 4b). The deviation is inevitable because the roughness of the edges affects the narrow linewidth. The ultrafine and smooth full metal nanostructures can be successfully fabricated by utilizing the template stripping (TS) method [31], which can create the ultranarrow linewidth in the experiment. Although the theoretically predicted ultranarrow linewidths have not been obtained by experimental means, the 12-nm linewidth experimentally presents a satisfactory result in the near-UV region.

## 4. Conclusions

In conclusion, we theoretically designed an Al nanoellipse metasurface located on the Al substrate. Importantly, the ultranarrow linewidth of 1.9 nm has been successfully obtained in the near-UV region. Simulated results demonstrate that such ultranarrow linewidths originate from the interaction of resonant modes when nanoparticles are arranged periodically. The important fact is that the ultranarrow linewidth (<2 nm) can be obtained by adjusting the periodic parameters in the DUV region. The proposed Al nanoellipse metasurface exhibits high sensing performance, achieving a FOM up to 211 nm/RIU. This theoretical research contributes to future exploration of the denaturation process of biomolecules in the UV and even the DUV regions, and additionally provides great potential for numerous applications, such as SERS and photoelectric detection in the UV or DUV regions.

## Figures and Tables

**Figure 1 nanomaterials-13-00092-f001:**
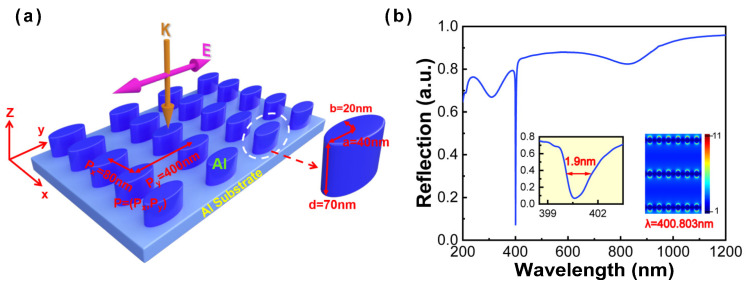
(**a**) Schematic diagram of the Al nanoellipse metasurface and the setting of structural parameters. (**b**) Reflectance spectrum of the Al nanoellipse metasurface with optimized parameters. The inset represents the zoomed-in image of the ultranarrow linewidth and near-field distribution of resonant wavelength corresponding to the ultranarrow linewidth.

**Figure 2 nanomaterials-13-00092-f002:**
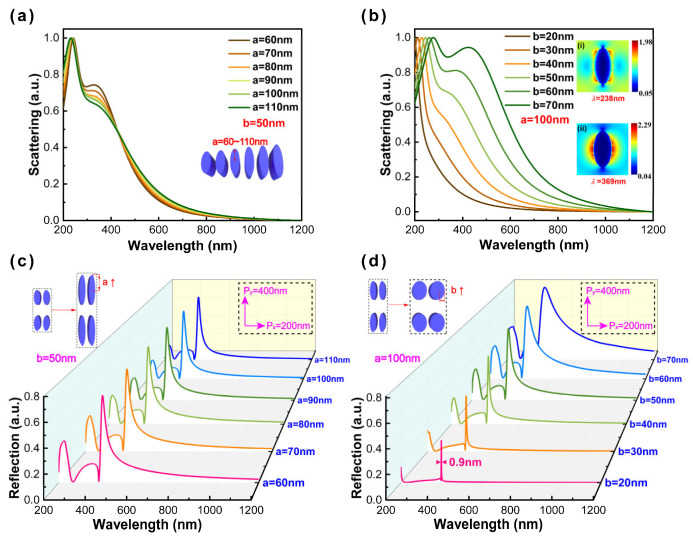
The far-field scattering spectra of individual nanoparticles with varied (**a**) semi-major axis parameters from 60 nm to 110 nm and (**b**) semi-minor axis parameters from 20 nm to 70 nm (at 10 nm intervals); the insets represent the near-field distribution corresponding to the wavelengths of 238 nm and 369 nm. The reflectance spectra of periodic arrays with varied (**c**) semi-major axis parameters from 60 nm to 110 nm and (**d**) semi-minor axis parameters from 20 nm to 70 nm (at 10 nm intervals). (The small window shows the schematic illustration of varied structural parameters of individual nanoparticles and periodic arrays).

**Figure 3 nanomaterials-13-00092-f003:**
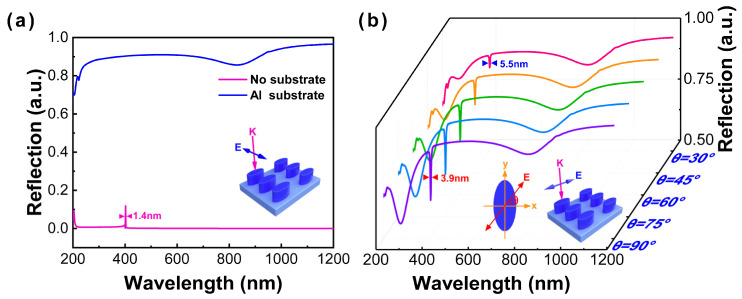
(**a**) Reflectance spectra of an Al nanoellipse metasurface on the Al substrate. (**b**) Reflectance spectra of an Al nanoellipse metasurface under different polarization angles from 30° to 90° corresponding to the pink solid line to the purple solid line.

**Figure 4 nanomaterials-13-00092-f004:**
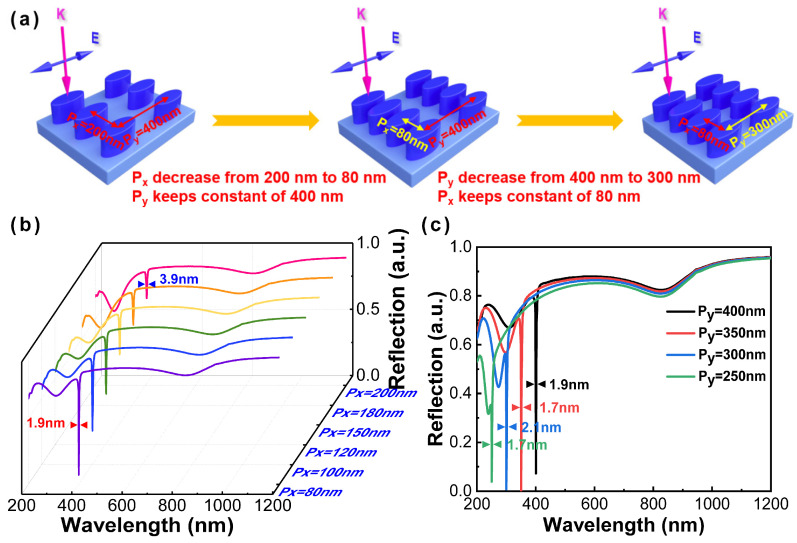
(**a**) Schematic illustration of periods of the Al nanoellipse metasurface. The corresponding reflectance spectra of the Al nanoellipse metasurface with varied (**b**) Px from 200 nm to 80 nm corresponding to the pink solid line to the purple solid line; (**c**) Py from 400 nm to 250 nm.

**Figure 5 nanomaterials-13-00092-f005:**
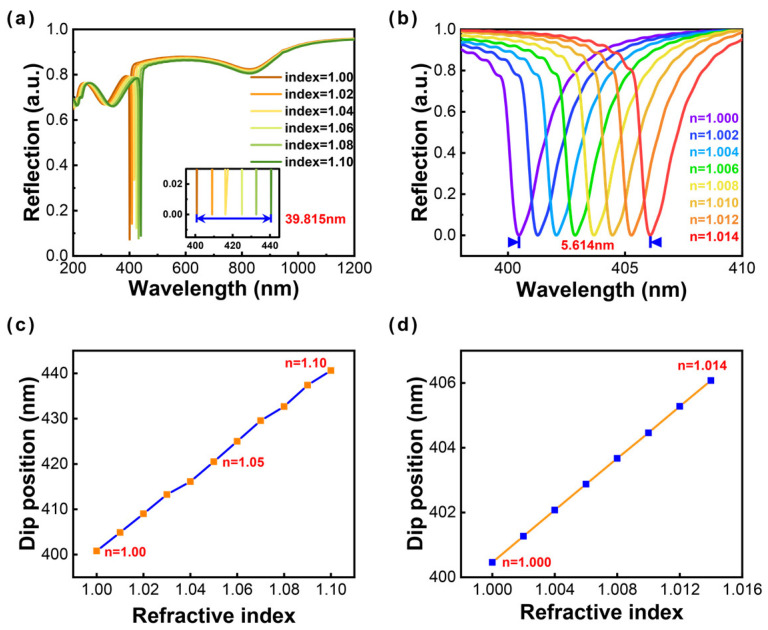
Reflectance spectra in different dielectric environments. The refractive index of the surrounding medium ranges (**a**) from 1.00 to 1.10 (the interval is 0.02); (**b**) from 1.000 to 1.014 (the interval is 0.002). The variation of dip position for two situations with the refractive index ranges (**c**) from 1.00 to 1.10 (at 0.01 intervals); (**d**) from 1.000 to 1.014 (at 0.002 intervals).

**Figure 6 nanomaterials-13-00092-f006:**
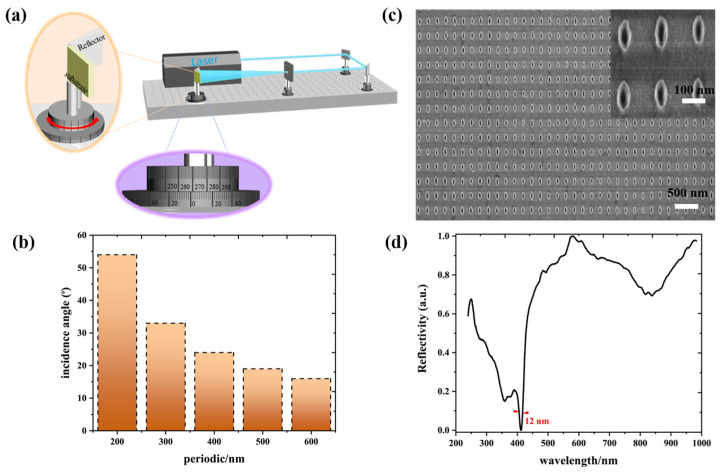
(**a**) Schematic of the LIL technology. Reprinted with permission from Ref. [23]. Copyright 2019 Royal Society of Chemistry. (**b**) The relationship between periods and incidence angles for wafer-scale plasmonic arrays. (**c**) Typical SEM image of a low-symmetry Al metasurface with 200 nm in Px and 400 nm in Py. (**d**) Reflectance spectra of a low-symmetry Al metasurface with 200 nm in Px and 400 nm in Py at normal illumination.

## Data Availability

Data underlying the results presented in this paper are not publicly available at this time but may be obtained from the authors upon reasonable request.

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
