# Peer review of "Extremely Ultranarrow Linewidth Based on Low-Symmetry Al Nanoellipse Metasurface"

_nanomaterials, 2022, doi:10.3390/nano13010092_

Round 1

Reviewer 1 Report

In the manuscript, authors theoretically designed aluminum nanoellipse metasurface. And the ultranarrow linewidth of 1.9 nm was successfully obtained in the near UV region. It could be expected the aluminum nanoellipse metasurface is useful for the plasmonic applications. The manuscript is well organized and the results are fine. However, the manuscript should be highly improved before considering of acceptance.

1. The mechanism to form ultranarrow linewidth is unclear, please give an explanation. For example, how does the large-range coupling occur when semi minor-axis is 20 nm? And 10 nm in semi minor axis may give a narrower linewidth?

2. Please also provide the near field distribution of resonance in the case of periodic array.

3. For experiments, please provide more detail. It is hard to understand how to obtain the periodic nanostructure based on the current description.

4. In experiment results, the nanostructure is periodic crater. While in the simulation, the nanostructure is nanoellipse. Why the authors compared the opposite nanostructures.

5. Generally, a thin layer of AlOx may exist on the Al substrate, especially after laser lithography, does the AlOx influence the optical properties of nanoellipse? And how to avoid the oxidation in laser lithography.

6. English writing must be edit by native person.

Author Response

Dear reviewers, We have carefully revised the manuscript according to reviewers’ detailed comments and suggestions. In the revised version of our manuscript, most of reviewers’ suggestions have been adopted. Some paragraphs were rewritten, and some new references and simulation data were added to address reviewers’ concerns. Our incorporation of comments from reviewers is given point by point in following separate sheets, namely, “Response and cover letter”. Please see the attachment.

Reviewer 2 Report

In this paper, metasurface with  nano elliptic pillars are investigated. The reflection spectrum for metasurface with different structural parameters are investigated in detail by FDTD method. Selecting structural parameters appropriately, ultra narrow linewidth is obtained. This metasurface can be used as a  refractive index measurement and the dependence of dip frequency on refractive index change is numerically demonstrated. This metasurface is actually fabricated and  the reflection spectrum is measured. The results may be interesting for the readers of this journal. On the other hand, I have some comments as follows:

1) The experimental results are little bit different from those by the numerical simulation. Especially, The reflectivity is greatly reduced in the low frequency side and linewidth is broaden. I understand that actual fabrication is difficult and the characteristic may be degraded. However, The dip shape is not clear compared with that obtained by numerical simulation.

2) The reflection spectrum as a function of refractive index change should be also shown by experimental measurement to demonstrate the results in Fig. 5. 

Author Response

(The authors gave the same response as above.)

Reviewer 3 Report

The manuscript (nanomaterials-2069042; Extremely ultranarrow linewidth down to 1.9 nm based on low-symmetry nanoellipse metasurface) describes the theoretical approach and experimental rationalization of ultranarrow linewidth metasurface. Theoretically and experimentally, 1.9 and 12 nm linewidth have been achieved. Because of the limitation of UV optical elements so far, SPR in UV regions become a more interesting topic in research society as a candidate for UV optical elements. Therefore, due to its importance, I cannot agree with a publication in this current form based on the issues below. In overall, a more dedicated and careful explanation is required.

Minor comments.

1.    Please use one of the following: Reflectance or Reflection.

2.    In Figure 2, from my guess, ‘a’ and ‘b’ seem to denote the long and short axis of ellipsoid. Please clarify the definition, not showing only sign on the figure.

3.    Schematic images in Figure 2, leads to confusing due to the perspective from rendering (same view-point of rendering). Please include a picture from the different view-points to make figure clear.

4.    The author stated that, Py is from 400 to 300 nm. But data shows 400 to 250 nm. Please correct the error.

5.    What does mean the “Scattering” in Figure 2? Is it a normalized cross-section or efficiency? Please provide a definition.

6.    Please clarify the description of the caption of Figure 6(b).

Major comments

1.    Please rationalize why the low-symmetry element is adopted in the metasurface. I cannot find any relevancy with the result.

2.    Figure 1 shows a sharp resonance at 400.803 nm. First, I would like to argue about the correctness of the value. The value would not be correct and looks like an approximation due to the numerical condition restriction. The value should depend on the number of frequency points in your FDTD domain-power monitor. Moreover, author should include field information from z-axis to clarify SPP excitation. The author may refer following articles:
(a) Wang, K., Schonbrun, E., Steinvurzel, P., & Crozier, K. B. (2011). Trapping and rotating nanoparticles using a plasmonic nano-tweezer with an integrated heat sink. Nature communications, 2(1), 1-6.
(b) Ding, F., Yang, Y., Deshpande, R. A., & Bozhevolnyi, S. I. (2018). A review of gap-surface plasmon metasurfaces: fundamentals and applications. Nanophotonics, 7(6), 1129-1156.)

3.    Simulation result analysis is not sufficient, the author should provide enough explanation of the result of at least the following:
(a) (Figure 3b) Ellipsoidal structural is very sensitive to polarization, but why bandwidth narrowing happens with a high polarization angle?
(b) (Figure 4b) Why Px reduction leads to bandwidth narrowing? Are there any blueshift of peaks?
(c) (Figure 4c) Why Py reduction leads to the blueshift of peak?

4.    I agree with there should be a disparity between experimental and theoretical approaches. However, the difference is way behind than expected. Data from Figure 6(d), reflection approaches 0 and it is even lower than the high-energy interband transition (UV region). Please explain the origin of the discrepancy between the experimental and theoretical approaches. In addition, if the experimental data is done with unpolarized light, the author should note the details.

5.    The linewidth of 1.9 nm is not the experimental value which may confuse readers. The use of the value in the title would be inappropriate. I recommend revising.

Author Response

(The authors gave the same response as above.)

Round 2

Reviewer 1 Report

The manuscript was highly improved. Then, I recommend to accept the manuscript.

Reviewer 3 Report

I am happy with the careful revision. I agree with publication.